# "I want to help my body": Acceptability of malaria chemoprevention among children with sickle cell anaemia and their caregivers in Malawi and Uganda

Sarah Svege[1]*, Siri Lange[2], Bjarne Robberstad[1], Joseph Rujumba[3]

1 Department of Global Public Health and Primary Care, University of Bergen, Bergen, Norway,
2 Department of Health Promotion and Development, University of Bergen, Bergen, Norway,
3 Department of Paediatrics and Child Health, School of Medicine, Makerere University, Kampala, Uganda

* sarah.svege@icloud.com

## Abstract

Patients with sickle cell anaemia (SCA) are at risk of severe illness and death if infected by malaria, and lifelong prophylaxis is recommended to individuals in malaria endemic regions. Although its efficacy is declining due to parasite resistance, the antimalarial drug Sulphadoxine-Pyrimethamine (SP) is still given to patients with SCA in several countries of sub-Saharan Africa. A clinical trial was performed to compare SP with Dihydroartemisinin-Piperaquine (DP) for malaria chemoprevention in children with SCA in Uganda and Malawi. This paper describes a study on acceptability which was nested within the trial. To explore views on malaria chemoprevention and the proposed treatment regimens, 29 focus group discussions were conducted with children above 10 years of age with SCA and caregivers of children with SCA. The discussions were transcribed and translated to English before coding and reflexive thematic analysis. Participants from the DP arm reported a reduced number of sick events and hospital admissions while they were in the trial, and ranked DP above SP in terms of perceived effectiveness. Although concerns were raised about initial side effects, a high pill burden, and the unpleasant smell and taste of tablets, most participants were willing to continue long-term administration of DP due to its observed or experienced health benefits. Despite positive attitudes towards the use of weekly DP, monthly dosing was frequently suggested as a better option as it would lower the pill burden and expand the time interval between treatment courses. To mitigate transport costs and time spent away from school and work, most participants preferred a period of two months or longer between drug refill visits at the hospital. During routine care visits, counselling about the importance of accurate dosing and ongoing adherence should be provided to ensure sustainable and successful use of malaria chemoprevention among children with SCA.

**Data availability statement:** Transcript excerpts which are relevant to the focus and content of this study are included in the text and tables as de-identified quotations. Although participants have been anonymised by assigning pseudonyms or participant numbers, the transcripts still carry identifiable information such as location data and demographic details. In addition, the complete transcripts contain sensitive information about trial participation and health care experiences at the various study sites. A public sharing of the transcribed material in its entirety would compromise the privacy of participants and study staff. To ensure data confidentiality and respect the ethical restrictions outlined in the research protocol and consent forms, the complete qualitative data set will not be publicly shared. However, extended excerpts may be shared upon request. The research protocol and consent forms have been approved by the School of Medicine Research and Ethics Committee in Uganda (REF 2020-103), the Uganda National Council for Science and Technology, College of Medicine Research Ethics Committee in Malawi (COMREC #19/11/2442), Regional Committees for Medical and Health Research Ethics in Norway (REK #30992), and the Institutional Review Board at Liverpool School of Tropical Medicine in England (LSTM REC 19-105). Queries regarding data access may be directed towards Professor Øystein Haaland at University of Bergen.

**Funding:** This work was funded by three sources: i) The Research Council of Norway through the Global Health and Vaccination Programme (GLOBVAC, project number 285284. GLOBVAC is part of the EDCTP2 programme supported by the European Union); ii) The Royal Norwegian Society of Sciences and Letters; and iii) The Faculty of Medicine at University of Bergen, Norway. The funders had no role in study design, data collection and analysis, decision to publish, or preparation of the manuscript.

**Competing interests:** The authors have declared that no competing interests exist.

## Introduction

More than half a million babies are born with sickle cell disease every year and at least 8 million people are currently living with this condition globally [1]. Nearly 80% of cases occur in countries of sub-Saharan Africa, and it has been estimated that 7.3% (4.03-10.57) of under-five deaths in Africa are caused by sickle cell disease [1–3]. Sickle cell disease denotes a group of genetic blood disorders, and sickle cell anaemia (SCA) is considered the most common and severe form [4]. Patients with SCA have two abnormal haemoglobin genes (HbSS) where one is inherited from each parent. Due to these mutations, their red blood cells adopt an abnormal, crescent-shaped configuration with shorter lifespans, reduced oxygen-carrying capacity and a tendency to cluster in blood clots [4]. When these clots hinder the flow of blood and oxygen to organs, muscles and bones, the patients experience episodes of excruciating pain known as 'sickle cell crises'. In addition to recurring pain attacks, they are at risk of organ dysfunction and cerebral strokes which may occur from early childhood [4].

The high frequency of haemoglobin mutations found in sub-Saharan Africa is linked to a process of natural selection where healthy carriers of *one* haemoglobin mutation, known as the 'sickle cell trait' (HbAS), have developed a degree of protection against malaria [5,6]. However, patients suffering from SCA with *two* haemoglobin mutations (HbSS) are not awarded the same evolutionary, selective advantage and are, on the contrary, at high risk of severe outcomes when infected [7,8]. Although there may be a lower overall incidence of malaria among individuals with SCA, several studies suggest that they are at increased risk of serious illness and death if admitted to hospital with malaria [7,9–12]. Over a decade ago, the World Health Organisation (WHO) recommended administration of long-term malaria chemoprevention to patients with sickle cell disease in malaria endemic areas [13]. Due to the impaired immunity associated with this condition, antimalarial drugs should be taken continually throughout the year irrespective of seasonal changes in malaria transmission. This strategy of continual malaria chemoprevention is tailored specifically to the high-risk group of children with SCA and differs from the intermittent administration of seasonal malaria chemoprevention which is recommended to young children during periods of peak transmission. Currently, there is no consensus on the most optimal drug of choice for children with SCA and countries of sub-Saharan Africa have adopted various chemoprevention regimens. In Nigeria, which accounts for a substantial number of annual SCA births and has the highest prevalence of malaria cases and malaria deaths worldwide [1,14], Proguanil is often administered to patients with SCA. However, its daily dosing schedule has proven challenging in terms of compliance with reported problems linked to the high pill burden and associated 'drug fatigue' [15,16]. In other countries of sub-Saharan Africa, such as Malawi and Uganda, monthly Sulphadoxine-Pyrimethamine (SP) is the current standard of care. Although monthly dosing may be more feasible for long-term use, there is a growing body of research indicating SPs declining efficacy as a result of parasite resistance [17–19]. This rise of resistance may be explained by the extensive use of SP for intermittent preventive treatment in children and pregnant women across

sub-Saharan Africa. Despite evidence of emerging resistance, SP is still given as malaria chemoprevention to patients with SCA in Malawi and Uganda, and there is now an urgent, unmet need for research on alternative avenues of antimalarial treatment.

Previous clinical trials have tested Dihydroartemisinin-Piperaquine (DP) in children and it has proven to be a long-acting, highly effective, and well-tolerated antimalarial drug with few side effects [20–22]. DP also provides an extended posttreatment prophylactic effect due to piperaquine's long terminal elimination half-life of 20–30 days [23]. However, apart from a study by Taylor and colleagues in Kenya [24], DP has not been widely tested in children with SCA. In the study by Taylor et al., monthly DP did *not* give a significant reduction in clinical malaria compared to either monthly Sulphadoxine-Pyrimethamine with Amodiaquine (SP-AQ) or the current standard of care in Kenya which is daily Proguanil [24]. Still, monthly DP was associated with a nearly 80% reduction in asymptomatic parasitaemia, no serious adverse events, and high levels of adherence and user acceptability [24]. Limited Plasmodium falciparum transmission and enhanced supportive care were listed as possible reasons for the low incidence of clinical malaria in this setting [24]. Similar studies in areas with higher rates of malaria transmission are now warranted to determine the most efficacious, safe, and acceptable strategy for malaria chemoprevention in this group of patients. A clinical trial was therefore performed by our research consortium in areas with moderate to intense malaria transmission in Malawi and Uganda [25]. The objective of this double-blind, randomised, multicentre, placebo-controlled, two-arm study design was to compare monthly SP with weekly DP for the **Chem**oprevention of **M**alaria in **Ch**ildren with Sickle Cell **A**naemia (**CHEMCHA**) in eastern (Uganda) and southern (Malawi) Africa.

### The CHEMCHA trial

The CHEMCHA trial was conducted in Uganda and Malawi from 2021 to 2023. In this trial, we found that weekly DP, relative to monthly SP, reduced the incidence of clinical malaria by 80% [25]. DP was well-tolerated and safe with a non-significant QT prolongation and 70% reduction in malaria-related hospital admissions. Despite a promising decline in malarious morbidity in the DP group, monthly SP was associated with fewer sick visits due to non-malaria illness, such as a lower likelihood of pneumonia and suspected sepsis [25]. Interestingly, the reduction in non-malaria sick visits in the SP arm was particularly pronounced among children above 5 years of age who were *not* given penicillin prophylaxis as part of standard care. It suggests that although SP is inferior in terms of its antimalarial abilities, it carries a considerable protective effect against other infections. This is likely related to the antibacterial and anti-inflammatory properties of the sulphadoxine component. However, in areas with high antifolate resistance towards SP, weekly single-day dosing of DP was a safer and more effective strategy for preventing malaria among children with SCA below 5 years of age who were receiving antibiotic prophylaxis [25]. In the case of children with SCA above 5 years of age who are *not* prescribed penicillin, DP should preferably be coupled with either an antibiotic or SP to protect against non-malaria related illness.

### The acceptability study

This paper presents findings from a study on acceptability which was nested within the CHEMCHA trial. The objective of this study is to explore views towards the antimalarial regimens from the perspective of patients and caregivers. Reported facilitators and barriers to uptake of long-term malaria chemoprevention will also be presented. When evaluating a health care intervention, an assessment of acceptability is instrumental as it may aid stakeholders and health personnel in their efforts to develop policy recommendations and successfully scale-up a new treatment. To ensure high levels of compliance and drug uptake, clinical evidence should be complemented by a consideration of the perceived benefits and barriers among patients and caregivers. Important insights from the receiver, end-user stance may enrich trial data by adding depth, detail, and nuance to the clinical findings.

There is a scarcity of studies describing malaria chemoprevention preferences among children with SCA and their caregivers, and to our knowledge this is the first study assessing acceptability towards *weekly* DP. In addition to a higher

protective efficacy against malaria, weekly DP may be superior to monthly DP in terms of safety with lower peak plasma piperaquine levels and, hence, a reduced risk of QT prolongation [25,26].

## Methods

### Study sites

Uganda is a country in East Africa with a population of 50 million, while Malawi is a South-Eastern African country with approximately 20 million inhabitants. Data collection for this qualitative study was performed at Jinja Regional Referral Hospital in eastern Uganda, Kitgum General Hospital in northern Uganda, and Kamuzu Central Hospital in Lilongwe, Malawi [25,27]. All hospitals have extensive catchment areas and are situated in regions with perennial patterns of moderate to intense malaria transmission.

### The malaria burden in Uganda and Malawi

In Uganda, malaria transmission is perennial with particularly intense rates of transmission during the two annual rainy seasons from approximately March to May and September to December. The malaria transmission in Malawi also follows a perennial pattern with its peak in the rainy season, which usually lasts from November until April. In 2023, the WHO African Region accounted for 94% (246 million) of malaria cases and 95% (569 000) of malaria-related deaths worldwide [14]. A total of 12 572 518 cases were recorded in Uganda, while 4 810 053 cases were identified in Malawi [14]. Children aged under 5 years represented 76% of all deaths in the WHO African Region [14]. Uganda is one of the countries with the highest malaria burdens globally [14]. Over half of all malaria cases in 2023 were attributed to Nigeria (25.9%), the Democratic Republic of the Congo (12.6%), Uganda (4.8%), Ethiopia (3.6%) and Mozambique (3.5%). Malawi, being a lesser country in terms of population, accounted for 1.8% of malaria cases in the region. Seasonal malaria chemoprevention with SP and Amodiaquine was implemented in Uganda for the first time in 2021 and 557 073 children have so far received at least one dose per cycle [14]. Unfortunately, we lack detailed information on the current coverage of seasonal malaria chemoprevention at the study sites in Uganda. Malawi has not yet implemented seasonal malaria chemoprevention, but it is one of the countries in a pilot implementation programme for the RTS,S vaccine against malaria. The RTS,S vaccine has proven to substantially reduce hospital admissions due to severe malaria [28]. A combination of malaria vaccines and seasonal malaria chemoprevention with SP and Amodiaquine lowered the incidence of severe malaria and death more than either of the interventions alone [29]. Although vaccines and seasonal malaria chemoprevention are likely to reduce the malaria-related disease burden among children, this study considers children with sickle cell anaemia who are supposed to receive continual, long-term malaria chemoprevention.

### Sickle cell disease in Uganda and Malawi

In the context of sickle cell disease, accurate assessments of country-specific disease burdens are hindered by a lack of newborn screening, late or absent diagnosis, and low public awareness about the condition [1,30–32]. Sickle cell disease is an umbrella term for all sickle cell disorders, and sickle cell anaemia (SCA) is considered the most severe and prevalent type [4]. One study suggests the prevalence of sickle cell trait and sickle cell disease in Malawi at 7,0% and 0,1%, respectively [33]. Estimates from Uganda indicate that at least 15 000 babies are born with sickle cell disease annually [34]. According to guidelines from the Ministries of Health in Malawi and Uganda, patients with sickle cell disease should be given monthly SP as part of routine care. However, the therapeutic threat of SP resistance has been identified in both countries [35–37], and this worrying development should spark the search for alternative antimalarial agents, such as DP, which may exert a higher degree of protection.

## Study population

A total of 724 children aged 6 months to 15 years were randomised to receive DP (n = 367) or SP (n = 357) for an average period of 15 months as part of the CHEMCHA trial. They were recruited during follow-up or sick visits at hospitals in Uganda (Jinja Regional Referral Hospital and Kitgum General Hospital) and Malawi (Kamuzu Central Hospital in Lilongwe and Queen Elizabeth Central Hospital in Blantyre). The inclusion criteria for enrolment were (1) age between 6 months and 15 years, (2) weight of 5 kg or more, (3) written informed consent from a parent or legal guardian coupled with an assent by the child if aged 8 years or older [25,27]. All children recruited to the trial had a diagnosis of SCA (HbSS) confirmed by either haemoglobin electrophoresis, high-performance liquid chromatography, or isoelectric focusing. For the qualitative study, participants were recruited from the CHEMCHA trial with purposive sampling of 70 children with SCA above 10 years of age and 117 caregivers of children with SCA below 10 years of age.

## Study design

The participants were blinded and randomly assigned to receive either monthly SP + weekly DP-placebo or weekly DP + monthly SP-placebo. This design ensured the same total number of tablets in the two study arms although some were placebo pills with no active ingredients. The participants were asked to attend hospital-based study clinics for drug refill, check-up, and adherence monitoring with residual pill counts every second month. All participants, caregivers, study staff, trial investigators, research assistants, and data analysts were masked to which treatment group the participants were assigned to during enrolment and the entire trial period. At the study clinics, the first dose of each treatment course was given with water under direct observation, while subsequent doses were administered to the child by a caregiver at home. In addition to the antimalarial treatment of their assigned study arm, the participants were given standard of care which includes daily folic acid, oral penicillin for children below 5 years of age, paracetamol for pain events, and updated immunisations [27]. A sub-sample of the study population (294/724) were taking hydroxyurea prior to enrolment, which was continued during participation in the trial [25]. Hydroxyurea is a disease-modifying drug that reduces the likelihood of red blood cell sickling by inducing the rise of foetal haemoglobin levels and inhibiting intracellular HbS polymerization [4]. This will, next, reduce the risk of blood cell sickling and pain episodes.

This paper describes a qualitative study which was performed as part of the CHEMCHA trial. A theoretical framework of acceptability by Sekhon and colleagues was applied to capture key dimensions of acceptability at different stages of the intervention [38]. Sekhon et al. define acceptability of a healthcare intervention as *"a multi-faceted construct that reflects the extent to which people delivering or receiving a healthcare intervention consider it to be appropriate, based on anticipated or experienced cognitive and emotional responses to the intervention"* [38]. In line with the selected framework, data collection was performed at three temporal phases: *early* in the enrolment stage in May 2022 (prospective acceptability), *during* active participation while the trial was ongoing in November 2022 (concurrent acceptability), and four months *after* exit from the trial in May 2023 (retrospective acceptability). Specifically, the time points were described as follows; (1) prospective acceptability in the pre-intervention period *prior* to any exposure to the intervention, (2) concurrent acceptability *during* intervention delivery when some exposure to the intervention has occurred, but there is still further exposure planned, and, finally, (3) retrospective acceptability *after* the intervention is completed and no further exposure is planned [38].

To increase the number of study participants represented in the FGDs, a new group of participants was recruited at each time point. Due to the blinded nature of the trial, the two first phases of data collection did not focus on drug-specific preferences, but rather operational elements of the intervention and attitudes towards long-term use of malaria chemoprevention. However, the final, retrospective phase was performed with a sub-sample of participants from the DP arm who, at that point, were unblinded and informed that they had been given DP in the trial. A data collection tool was developed for each phase and all tools were carefully reviewed and agreed upon by the authors prior to use. The tools comprised of open-ended questions on malaria chemoprevention, preferred treatment intervals and drug delivery. A few hypothetical

scenarios on implementation were included to elicit the participants' thoughts and concerns about the feasibility and scalability of the intervention outside of a trial setting.

## Data collection

A total of 29 focus group discussions (FGDs) were carried out with 187 participants to capture and compare a wide set of views and preferences. FGDs were deemed appropriate in this setting as the topics of interest, such as treatment preferences and views on malaria chemoprevention, are not very sensitive in character [39,40]. In addition, FGDs allow for sharing and comparing of thoughts and experiences among participants, as well as observation of group dynamics and interactions between participants [39]. The discussions lasted for 1 ½ - 2 hours and mainly consisted of 6 participants, although some had a lower (4–5) or higher (7–8) number of participants. Recruitment was done through phone calls performed by the CHEMCHA study staff. However, the study staff was not present at the FGDs. All participants received travel reimbursement and refreshments. The FGDs were moderated by a team of female (n = 3) and male (n = 3) research assistants (RAs) with extensive experience in qualitative data collection. The RAs were fluent in the local languages at the sites where they worked: Chichewa (Lilongwe, Malawi), Luganda or Lusoga (Jinja, Uganda), and Luo (Kitgum, Uganda). One or two of the authors were present at all group discussions. Daily debriefing sessions were organised to discuss emerging insights and plan for subsequent data collection.

## Data analysis

All FGDs were audio recorded, transcribed verbatim, and translated to English. Transcripts were exported to the NVivo software programme version 12 for coding. The reflexive thematic analysis approach by Braun and Clarke was applied to identify important patterns of meaning within the dataset [41]. The following constructs from the framework by Sekhon et al. were used to evaluate acceptability: *affective attitude, burden, self-efficacy, intervention coherence, perceived effectiveness, opportunity costs, and ethicality* [38]. These constructs served as the main, top-level codes in the coding frame. Although the codes were mainly theory-driven and deductive, the process of analysis still demanded careful and critical engagement with the dataset to determine how the participant statements should be appropriately situated in the framework. When assigning transcribed material to various constructs, the analysis oscillated between a semantic, surface-level investigation and a more detailed dive to discover implicit, deeper layers of meaning. The analysis resulted in five overarching themes where each theme is comprised of codes with interlinked issues of interest. The demographic characteristics are drawn from questionnaire data obtained prior to the FGDs, and calculations of the numerical values in % and n were done in Microsoft Excel.

## Ethical considerations

Ethical clearance was obtained from the School of Medicine Research and Ethics Committee in Uganda (REF 2020-103) and the Uganda National Council for Science and Technology, College of Medicine Research Ethics Committee in Malawi (COMREC #19/11/2442), Regional Committees for Medical and Health Research Ethics in Norway (REK #30992), and the Institutional Review Board at Liverpool School of Tropical Medicine in England (LSTM REC 19-105). A written informed consent was obtained from all study participants prior to the FGD. For children, a separate assent form was completed in addition to the parental consent form. The consent form was read to or by the participant before signing and consisted of information about confidentiality, rights to withdrawal, and the purpose of the study. In cases of low literacy, a thumbprint was used to document consent and the consent form was read to participants in the presence of a literate impartial witness. To ensure confidentiality, all participants were assigned anonymised names or numbers in questionnaires and transcripts. Consent forms with signatures and identifiable information were kept separate from transcripts in a lockable study office which can only be accessed by the authors. Similar procedures of consenting, anonymising, and

PLOS Global Public Health

data storage were adhered to at all study sites. Additional information regarding the ethical, cultural, and scientific considerations specific to inclusivity in global research is included in the S1 Checklist.

## Results

This section presents the demographic characteristics of participants followed by five themes derived from the analysis. The themes explore the participants' cognitive and emotional responses towards various malaria chemoprevention strategies.

### Demographic characteristics

A total of 70 children with SCA above the age of 10 years took part in the FGDs. The number of FGDs conducted at each phase and study site is shown in Table 1. Most of the children were 10–12 years of age (38%) and had completed 5–7 grades of schooling (51.4%). In addition, 117 caregivers of children below 10 years of age with SCA took part in FGDs. The vast majority of caregivers were female (89.7%), married (83.8%), and stated farming (42.7%) or business and sales (33.3%) as their occupation. An overview of demographic characteristics can be found in Tables 2 and 3.

### Preferred treatment interval: "I choose monthly because for humans too much of anything becomes monotonous"

Across three temporal stages of intervention delivery and three study sites, most caregivers and children viewed *monthly* administration of antimalarial drugs as more feasible and favourable than *weekly* administration. Monthly

**Table 1. Overview of data collection phases and focus group discussions (FGDs).**

| Study phase | Prospective | Concurrent | Retrospective | Total |
|---|---|---|---|---|
| Study sites | Kitgum<br>Jinja | Kitgum<br>Jinja<br>Lilongwe | Kitgum<br>Jinja | 3 |
| Study countries | Uganda | Uganda<br>Malawi | Uganda | 2 |
| FGDs with caregivers | 4 | 11 | 3 | 18 |
| FGDs with children | 2 | 6 | 3 | 11 |
| Total (FGDs) | 6 | 17 | 6 | 29 |

**Table 2. Demographic characteristics of children (n = 70) in n and %.**

| Variable | No (%) |
|---|---|
| **Gender** | |
| Male | 36 (51.4) |
| Female | 34 (48.6) |
| **Age** | |
| 10–12 | 38 (54.3) |
| 13–14 | 22 (31.4) |
| ≥15 | 10 (14.3) |
| **Education (grades)** | |
| 1–4 | 27 (38.6) |
| 5–7 | 36 (51.4) |
| >7 | 7 (10.0) |

**Table 3. Demographic characteristics of caregivers (n = 117) in n and %.**

| Variable | No (%) |
|---|---|
| **Gender** | |
| Female | 105 (89.7) |
| Male | 12 (10.3) |
| **Age group** | |
| 20–30 | 39 (33.3) |
| 31–45 | 65 (55.6) |
| >45 | 13 (11.1) |
| **Education** | |
| Primary | 44 (37.6) |
| Secondary | 45 (38.5) |
| Tertiary | 12 (10.2) |
| None | 16 (13.7) |
| **Marital status** | |
| Married | 98 (83.8) |
| Single | 19 (16.2) |
| **Occupation** | |
| Farmer | 50 (42.7) |
| Business/trader | 39 (33.3) |
| Home-keeper | 16 (13.7) |
| **Religion** | |
| Christian | 99 (84.6) |
| Muslim | 18 (15.4) |
| **Number of children** | |
| 1–2 | 29 (24.8) |
| 3–4 | 48 (41.0) |
| ≥5 | 40 (34.2) |
| **Number of children with sickle cell anaemia** | |
| 1 | 84 (71.8) |
| 2 | 29 (24.8) |
| 3 | 4 (3.4) |

regimens were linked to a lower pill burden and less time spent on preparing, performing, and monitoring drug administration. If the children experienced drug-related discomfort such as nausea and vomiting or complained about an unpleasant smell and taste of tablets, they and their caregivers were more likely to prefer monthly instead of weekly treatment courses. It was reported how children may resist taking the drugs and request food or juice to ease or eliminate the smell and taste of the drug. In such instances, the caregivers felt inclined to support monthly regimens with longer drug-free intervals, referred to as 'rest periods', between doses. Monthly treatment was also associated with a perceived reduced risk of 'drug fatigue' among children. When asked to share their treatment interval of choice, most participants would opt for monthly doses:

> "Monthly is better since every time a child sees the drug after one whole month, it will be like a surprise. He can even say: 'although I am on this drug for a long time, the time interval between when I have to take it is longer', but for the weekly one, he may get demoralised, develop a fear and wonder: 'for how long am I going to take this drug?' (Mother, Jinja)."

Still, a few participants argued that weekly drugs may provide better protection against malaria compared to monthly drugs:

*"Somehow I feel the weekly dosing has more benefits than monthly because it is a shorter interval between each time we are giving the drugs, and therefore it is more effective in preventing malaria from infecting the children (Father, Kitgum)."*

Also, some suggested that a once-weekly treatment day is easier to remember since it may be linked to specific weekly events such as days for prayer, the use of sports clothing to school, and a specific TV or radio show.

### Preferred drug refill site: "We should continue coming to the hospital because if the child has any other sickness, the health worker will be able to see it"

Most participants preferred to access the hospital for drug collection. In their view, the drugs would be more safely stored at the hospital compared to local health centres. A reported added value of drug collection at the hospital was continuing contact with experienced health workers who may perform clinical check-ups and provide treatment if necessary. Due to difficulties with securing sufficient funds for hospital transport, an extended time interval of two months or more between each drug refill visit was preferred across all sites. Such longer time intervals were frequently suggested by caregivers as it would lessen their time spent away from work and farming. In addition, it would minimise absence from school which was raised as a concern by the children:

*"For me it was challenging because it is a long distance from my school to the hospital, so transport was difficult and during exams you will have to miss it. Secondly, even if it is not exams, you still have to leave school to refill the drugs while others would have moved forward in class and you will be left behind (Child, 12 years old, Kitgum)."*

Participants at all three study sites detailed the struggles and setbacks associated with meeting the costs for hospital transport. To mitigate travel time and expenses, a few participants spoke in favour of drug delivery at local health centres instead of the hospital.

### Preferred antimalarial drug: "Our children were taking SP before, but they would still get sick as if they were not on medication"

When evaluating concurrent acceptability while the trial was still ongoing, it was evident that multiple caregivers had noticed an improvement in the health and physical appearance of their children after taking part in the trial, with one mother describing the following:

*"The study has helped the appearance of my child. The child now looks stronger and the eyes are not as yellow as they would get sometimes. The child even noticed and said: 'mother, with these drugs I feel different. Before when you asked me how I was doing, I would lie and say I was fine and yet my body felt heavy and I was not okay at all, but these days I wake up feeling energetic' (Mother, Lilongwe)."*

Several caregivers stated that they would now visit the hospital only for drug collection, while in the past their children were more frequently admitted due to episodes of ill-health. During the post-trial FGDs, unblinded participants from the DP arm reported similar patterns with less or no malaria-related hospitalisations when they were enrolled in the trial. This reduced rate or lack of hospital admissions was partly attributed to the DP drug and, thus, served as evidence of the drug's 'strength'. The retrospective phase of data collection also revealed that these health benefits were seemingly less apparent *after* trial exit when they returned to the standard of care (monthly SP):

*"On my part, I have a lot of challenges - even right now we are admitted in the hospital because my child has malaria. When we were in CHEMCHA, the drugs the child was taking were helping, but now the malaria episodes have increased which means that the medicine we used before (DP) worked better than the one (SP) we are taking now (Mother, Jinja)."*

During the FGDs with children, it was reported that the care received in CHEMCHA improved their energy level, school attendance, and ability to play and interact with friends. Many children in the DP arm attributed their strengthened health and wellbeing to the DP drug. However, DP was also linked to an unpleasant smell and taste, and side effects such as nausea, headache, and dizziness. In most cases, the side effects occurred during the initial treatment courses and would subside as the study progressed. The perceived benefits of the study drugs and trust towards the health workers seemingly strengthened as side effects reduced, the number of sick events declined, and they were offered close follow-up care in the ensuing months. This illustrates the dynamic nature of acceptability where positive attitudes towards the intervention and study drugs would grow in prominence and prevalence throughout the trial period.

Irrespective of initial complaints about side effects and the disadvantage of high pill numbers in older children due to weight-dependent dosing, most participants were committed to continue with DP if it is proven to be more effective than SP. In a similar vein, several caregivers expressed a willingness to comply with whichever dosing schedule and quanta of pills deemed necessary to achieve adequate protection against malaria. Memories of past sick events and hospitalisations were highlighted by caregivers as important drivers for the continued use of antimalarial drugs. Hence, their motivation to follow healthcare instructions was fuelled by a fear of reliving past experiences where the child's condition was deteriorating:

*"I think if it is discovered that DP which we take weekly is better and it is the one we are going to use, I would sit down with the child and explain: 'let us swallow this drug – this is what is going to save us and avoid malaria so that you may feel fine', and I will make sure that the child takes the drugs for as long as it is necessary because I could never forget how the child used to get pain attacks before (Mother, Kitgum)."*

**Views on long-term malaria chemoprevention: "I have now accepted it because my life is dependent on drugs"**

Children in multiple FGDs considered their lives to be 'dependent on drugs' and acknowledged the importance of taking drugs to prevent malaria:

*"For me, I have now accepted it because my life is dependent on drugs (Child, 11 years, Jinja)."*

Their willingness to continue with malaria chemoprevention was often rooted in the notion that antimalarials, and other drugs of preventive nature, may shield them from serious infections and sustain their health and wellbeing. Also, several children stated that treatment recommendations should be followed to achieve one's dreams and aspirations. Still, when asked to reflect around a scenario where antimalarial drugs are supposed to be taken for the rest of their lives, contrasting views emerged:

*"I can take the drugs for the rest of my life since it is protecting my life. If the hospital says that we should be taking the medication and you choose not to take it and die then you have only yourself to blame and not the hospital (Child, 13 years old, Lilongwe)."*

*"Taking these drugs is hard and sometimes I honestly feel like throwing them away. So, if you say that we have to take them for the rest of our lives; some of us will be throwing them away while others may be taking them. For example, myself, I cannot be taking drugs for the rest of my life (Child, 11 years, Lilongwe)."*

 

A few children and caregivers were worried about the negative effects of taking drugs for a long time. Some argued whether long-term use may lead the body to gradually adapt and become immune to the drug's preventive powers. One mother was concerned about the impact of long-term drugs on her child's organs:

*"I do not feel good about my child taking drugs for the rest of his life because when someone takes drugs for a very long time it can damage the liver and other parts of the body - but if these drugs are preventing the worst from happening then we have no option but to accept it (Mother, Jinja)."*

Also, some caregivers feared that their children may develop 'drug fatigue' as they grow older and resist, forget, or stop taking the drugs as instructed. To ensure ongoing compliance, the value of counselling and creating a 'common ground' with their children was emphasised:

*"I believe that the children will be able to take these drugs because I have seen it with mine. There was someone who told him: 'this one is going to die. He is always taking drugs'. But my son answered them by saying: 'I am not going to die. I am going to live and I am going to take these drugs and I will live'. So he has that courage and those around him were amazed. However, we have to counsel the children and make them understand that if they take this medicine they will live, but if they don't, they will encounter problems (Mother, Jinja)."*

The reported benefits and barriers of various malaria chemoprevention strategies are summarised in Table 4.

**Views on the CHEMCHA study: "They picked us out of the ditch and placed us on the surface"**

At the concurrent and retrospective stages of intervention delivery, a high degree of user satisfaction towards the treatment programme in CHEMCHA was identified. A sentiment of appreciation permeated through numerous accounts at the three study sites. The participants were particularly grateful to the CHEMCHA health workers who were commonly characterised as caring, skilled, and attentive with hearts resembling those of a friend or parent:

*"Whoever you approach here has the heart of a parent and you would think they were all born by the same woman. For the one and a half years we have been in CHEMCHA, they have taken very good care of us, and we feel relieved in our hearts because they picked us out of the ditch and placed us on the surface (Mother, Jinja)."*

This narrative depicts how CHEMCHA study staff were a source of relief and enabled the caregivers to overcome some of the struggles they faced when caring for their children during the trial period. It was repeatedly stated by caregivers how close follow-up, constant access to drugs, and travel support in the trial granted them some long-awaited relief from the financial stressors and uncertainties of their usual healthcare-seeking endeavours. During the FGDs held in the concurrent phase while the trial was still ongoing, many parents expressed that they were apprehensive about the upcoming return to their pre-trial realities, and one father shared the following:

*"I fear what may happen in the future. I know that since this is a study, there will come a time when it has to end. But I believe that the government listens to the voices of the people who conducted the study and I pray that you give a strong recommendation. Be our voice! You are like water drops that started raining and afterwards disappear leaving the crops here for the sun to scotch them - meaning that you brought us to this point in a very good way and now you are leaving us. If you leave us like this, the study will be meaningless because we even reached a point where we believed our children could survive (Father, Kitgum)."*

**Table 4. Reported benefits and barriers of various malaria chemoprevention strategies.**

| Topic of interest | Benefits | Barriers |
|---|---|---|
| Weekly treatment intervals of antimalarial drugs | • Perceived better protection against malaria than monthly regimens<br>• Higher preventive effect associated with more frequent treatment courses<br>• Easier to remember weekly drugs than monthly drugs | • Increased pill burden with weekly treatment intervals compared to monthly treatment intervals<br>• A higher risk of 'drug fatigue' among patients<br>• The perceived negative effect of long-term drugs on organs<br>• Limited 'rest time' between treatment courses |
| Drug refill visits at the hospital every second month | • Safe storage and perceived better durability of drugs at hospital pharmacy than local health centres<br>• Drug refill in the presence of skilled and experienced health workers at the hospital<br>• Drug collection at the hospital may be combined with clinical check-ups and sickle cell routine care | • Absence from school and work<br>• Long travel distance to hospital<br>• Lack of travel funds |
| Dihydroartemisinin-piperaquine (DP) as malaria chemoprevention | • Less malaria episodes<br>• Less hospital admissions<br>• Improved energy and appetite<br>• Better school attendance<br>• 'Encouraged by experience:' willingness to comply due to experienced benefits | • Unpleasant smell<br>• Bitter taste<br>• Initial side effects such as nausea, headache, and dizziness |

Among those interviewed in the post-intervention, retrospective phase, there was a widespread wish for the study to resume its activities, and the requests for continued support were uttered with a palpable sense of urgency. An overview of significant statements and thematic trends connected to various acceptability constructs is provided in Table 5.

## Discussion

If a healthcare intervention, such as the antimalarial regimen proposed in CHEMCHA, is considered acceptable, patients are more likely to adhere and benefit from improved clinical outcomes [38]. A high degree of acceptability towards DP as malaria chemoprevention has previously been identified among caregivers of children with SCA in Kenya and caregivers of children *without* SCA in Malawi [24,42]. However, in these studies DP was administered at *monthly* intervals and for a shorter duration compared to the CHEMCHA trial. The proposed weekly regimens of single-day DP also differ from the three-day treatment courses in prior studies [24,42].

To our knowledge, this is the first study assessing acceptability towards *weekly* single-day treatment courses of DP. Previous studies have predicted weekly dosing of DP to be more effective at preventing malaria episodes than monthly dosing [26,27]. Weekly dosing has been associated with a lower selection pressure for resistance, less safety concerns pertaining to peak piperaquine levels, and fewer consequences of missing occasional doses [26,27].

In the final phase of data collection, we recruited a sub-sample of participants who had received weekly DP for an average period of 15 months as part of the parent trial. The retrospective phase was performed *after* trial finalisation and unblinding. At this point, the participants were supposed to revert to monthly SP, which allowed them and their caregivers to apply a comparative lens when evaluating the standard of care (SP) versus the proposed drug regimen (DP). Most study participants ranked DP above SP in terms of its antimalarial properties and health benefits. Still, some asked whether DP may be formulated in such a way that monthly administration would be sufficient to attain the desired level of protection. The concern raised about weekly dosing of DP was related to the higher pill burden and increased risk of 'drug fatigue'. Thus, monthly DP was suggested by some participants as the best strategy for achieving long-term adherence.

**Table 5. Summary of acceptability constructs, significant statements, and thematic trends.**

| Acceptability constructs and significant statements | Conceptual definition of constructs | Thematic trends identified within the dataset |
|---|---|---|
| AFFECTIVE ATTITUDE<br><br>*«I am so grateful because I found peace in this programme and they have actually given me hope that my child will grow»*<br>*(Mother, Lilongwe)* | How an individual *feels* about the intervention (**affective attitude**) | • Gratitude towards health staff<br>• Fear of study finalisation<br>• A feeling of relief due to drug availability and transport support in CHEMCHA |
| BURDEN AND SELF-EFFICACY<br><br>*«It was hard for me to take those drugs (DP) because they smell bad and during my first time taking them I got a headache, dizziness and would feel like vomiting. The other thing is that the pills are so many»*<br>*(Child, 12 years, Jinja)*<br>*«For me, I do not have any fears because I want to help my body»* *(Child, 13 years, Jinja)* | The perceived amount of effort required to participate in the intervention (**burden**)<br><br>The participants' confidence that they can perform the behaviour(s) required to participate in the intervention (**self-efficacy**) | • High pill burden<br>• Bad taste and smell of drugs<br>• Initial side effects<br>• Frequent study visits<br>• Willingness to comply with healthcare instructions<br>• Motivation to continue with antimalarial drugs after observing health benefits<br>• Health is 'dependent on drugs'<br>• Anticipated 'drug fatigue' and compliance issues as children grow older |
| INTERVENTION COHERENCE AND PERCEIVED EFFECTIVENESS<br><br>*«They want to compare monthly and weekly drugs to see which one can help the children the best. We cannot tell which drug is working better, but we can see that it is doing good because our children are now growing well»*<br>*(Mother, Kitgum)* | The extent to which the participant understands the intervention and how it works (**intervention coherence**)<br><br>The extent to which the intervention is perceived as likely to achieve its purpose (**perceived effectiveness**) | • Accurate description of study rationale and treatment intervals<br>• Confusion about placebo pills<br>• Less malaria episodes<br>• Improved health and wellbeing<br>• Reported increase in energy level, appetite, growth, and school attendance<br>• Perceived decline in the drug's protective ability with long-term use |
| OPPORTUNITY COSTS AND ETHICALITY<br><br>*«My child has taken all sorts of drugs and if he was not taking drugs, he would be dead a long time ago. There are times when you say to yourself; should I leave him and not give the drugs today? But with the current situation you know that you have to continue even if he likes it or not»* *(Mother, Jinja)* | The extent to which benefits, profits, or values must be given up to engage in the intervention (**opportunity costs**)<br><br>The extent to which the intervention has good fit with an individual's value system (**ethicality**) | • Absence from school and work during study visits<br>• Transport costs and travel time<br>• Easy transition to added study drugs in families accustomed to regular drugs as part of routine sickle cell care<br>• The importance of caregiver encouragement and counselling for continued compliance<br>• Perceived negative effect of drugs on organs |

Prior studies have indicated a high level of acceptability towards malaria preventive treatment given as part of routine care in pregnancy and through the expanded programme on immunization for newborns [43–45]. Administration of seasonal malaria chemoprevention to young children and intermittent preventive treatment in school-aged children have also proven to be highly accepted among parents [46,47]. It is likely that certain features of these chemoprevention strategies, such as the treatment duration being restricted to a specific age group or time period, enhanced the level of acceptability. However, temporary, short-term treatment is not applicable in the context of a chronic condition such as SCA where the impaired immunity and risk of complications entail a continuous, life-long provision of prophylaxis. As opposed to malaria chemoprevention given to infants in their first year of life or children during peak transmission, patients with SCA require long-term administration of antimalarial drugs. The reported views towards malaria chemoprevention in this study imply that it was wellaccepted by most participants. Although they shared a willingness to follow the recommended regimens, one may still expect that long-term adherence and acceptance could be threatened by the indefinite duration of treatment. Thus, successful and sustainable uptake of malaria chemoprevention should go hand in hand with ongoing counselling about the value of drugs given for preventive purposes and the importance of continued compliance.

There is ample evidence attesting to the frequent confusion about malaria medicines given for 'protection' rather than 'treatment', and the value of giving drugs to healthy children without symptoms of malaria has been questioned by caregivers [48–50]. Previous studies have highlighted how caregivers may stop giving antimalarial drugs if the child appears healthy and, rather, 'save' remaining doses for future episodes of illness [51–53]. However, non-adherence or underdosing should be avoided as it may encourage the emergence of drug resistant parasites and slow the progress of malaria control and elimination efforts [54–56]. As opposed to the more immediate effects of certain curative drugs, which aim at relieving patients from pain and discomfort, the clinical benefits of preventive drugs are less apparent. For example, the direct impact of antimalarial drugs is not instantly felt or observed as their pharmacological effects are executed over time with the purpose of preventing progression of 'hidden' or 'silent' asymptomatic parasitaemia to actual signs and symptoms of disease [49,53,57].

In the context of malaria prophylaxis, acceptability studies have generally focused on intermittent preventive treatment during infancy and pregnancy, seasonal malaria chemoprevention, or mass drug administration [57]. These malaria preventive strategies are provided at population level in malaria endemic areas irrespective of the clinical condition of its recipients [46,58]. However, this study addresses acceptability of malaria chemoprevention tailored specifically to patients with SCA in malaria endemic regions. Patients with SCA are chronically ill with frequent pain episodes and complications which may be further exacerbated by malaria. In a high number of FGDs, the participants were acutely aware that malaria may worsen the condition of children with SCA. This knowledge seemingly strengthened their commitment to continue long-term administration of antimalarial drugs. The chronic, severe nature of SCA coupled with a widespread fear of pain attacks and complications could, thus, be regarded as enabling factors for treatment uptake. This echoes a previous study from Uganda where the severity of SCA was identified as one of the major drivers for adherence to disease-modifying medications [59].

Despite the reported logistical and financial challenges of hospital transport, most participants preferred drug collection at the hospital instead of local health facilities. There was a general agreement that drugs should be safely stored at the hospital. According to several participants, drug refill visits at the hospital would, to a greater extent than at local health centres, allow access to quality care and support from experienced health workers. In terms of treatment intervals, monthly administration was favoured by many due to a lower pill burden than weekly intervals and the perceived reduced risk of anticipated 'drug fatigue' among children. Initial side effects, unpleasant smell and taste of tablets, and concerns about the negative impact of drugs on organs were expressed as possible barriers to the use of DP. Although side effects have previously been interpreted as 'evidence' of a drug's potency and power [60,61], they are, as our study suggests, most commonly considered a barrier to uptake [51,62]. Some participants also raised concerns about whether the protective ability of drugs may lessen with long-term use, and in this era of emerging resistance such worries are to some extent rooted in reason and valid scepticism. Still, for children with a life-threatening condition such as SCA, it is likely that the overall clinical benefits of malaria chemoprevention will outweigh the downsides of initial, minor side effects and the potential negative impact of drug exposure on organs and parasite resistance. This element of expected risk versus reward should be clearly communicated to patients and caregivers during routine care visits.

During the first stages of intervention delivery, there was some scepticism concerning the safety and side effects of the drugs. However, this reluctance seemingly subsided as the initial side effects disappeared, familiarisation with the study procedures and personnel increased, and the health benefits of the children became more evident. The retrospective phase of data collection revealed a particularly high level of user satisfaction towards the intervention and a wish for continued care and provision of drugs. Performing data collection at various phases of an intervention exemplifies how extended exposure to a treatment, such as weekly DP, may strengthen the degree of acceptability among participants.

Participants in the DP arm reported improved health with less malaria episodes and fewer hospital admissions. However, several caregivers claimed that the health of their children regressed after trial exit. Although an actual decline in the health state of their children may have occurred post-trial, it is also possible that the experienced change in 'caregiving

security' with loss of safe drug access and travel support prompted these perceptions. When the families were funnelled back into the uncertainties of their 'everyday lives' outside of a trial context, the highly feared, but familiar, care-seeking concerns and barriers such as transport difficulties, long waiting time, drug stock-outs, and treatment costs made a dreaded return. Thus, the recurring reports about enhanced health and wellbeing were likely coloured by the comprehensive care and support provided in the trial. The disease-modifying drug hydroxyurea was administered to a considerable proportion of the study population and may also have contributed to the reduction in pain crises and hospitalisations among these patients. The reports about positive health effects should, thus, not be attributed solely to the prescribed antimalarial drugs, but rather to the trial context more broadly. However, concerns raised about the high pill burden and feared 'drug fatigue' among children would presumably be less prevalent in a non-trial setting *without* added placebo.

The serious, chronic nature of SCA demands diligence in matters of care and compliance, and most families in this study were already accustomed to multiple daily or regular drugs such as folic acid, penicillin, and hydroxyurea. Thus, they did not consider the addition of study drugs in CHEMCHA as a particularly abrupt transition in drug burden nor treatment behaviour, which may also explain the high level of acceptance found in this setting. It is recommended that children with SCA attend regular follow-up visits for clinical check-up and collection of medication. These routine care visits offer a crucial point of contact for delivery of drugs and repeated counselling about the importance of ongoing adherence to malaria chemoprevention. Apart from a higher number of tablets with weekly and weight-dependent dosing, provision of DP as malaria chemoprevention would not entail a significant shift in terms of required resources in settings where routine care practices are already integrated within the existing health system.

### Study limitations

The reported findings are drawn from a trial context and do not reflect a 'real-life setting' which is commonly constrained by low availability of drugs, medical equipment, and health personnel. The participants in CHEMCHA received close follow-up, travel reimbursement, and safe access to drugs. It is likely that this trial context encouraged their predominantly positive attitudes towards the intervention. In the two first phases of data collection, positive reports about health improvement were difficult to attribute to either SP or DP because the participants were still blinded. However, the final phase of data collection allowed us to, after unblinding, retrospectively assess acceptability towards DP. Malawi is one of the countries in the WHO-coordinated Malaria Vaccine Implementation Programme, where 6 million doses were delivered to 2 million children in Ghana, Kenya and Malawi between 2019 and 2023. An evaluation of impact identified a 13% reduction in all-cause mortality and a 22% reduction in hospitalisations for severe malaria [14]. Malaria vaccines are currently being rolled out in several countries of sub-Saharan Africa, including Malawi. At present, it is still uncertain how this will affect the prevalence of malaria among children with SCA and whether the vaccine should be administered alone or in combination with malaria prophylaxis in this group of patients.

### Conclusion

This study revealed a high degree of acceptability towards malaria chemoprevention among children with sickle cell anaemia and their caregivers in Malawi and Uganda. A willingness to comply with treatment guidelines and continue long-term malaria chemoprevention was apparent across three study sites. To facilitate long-term adherence, most participants preferred monthly treatment intervals and a period of two months or longer between each drug refill visit at the hospital to reduce travel costs and time away from school for children and income-generating work for caregivers. Children who received weekly Dihydroartemisinin-Piperaquine for an average period of 15 months as part of a clinical trial, entitled CHEMCHA, experienced an improvement in their energy level, school attendance, and overall wellbeing. The caregivers of children who were given Dihydroartemisinin-Piperaquine reported a decrease in sick events among their children with less malaria episodes and fewer hospital admissions. Due to its clinical benefits, most participants favoured Dihydroartemisinin-Piperaquine over the standard of care Sulphadoxine-Pyrimethamine for prevention of malaria. During

routine care visits, children with sickle cell anaemia and their caregivers should be provided counselling about the importance of ongoing adherence to achieve optimal protection against malaria.

## Supporting information

**S1 Checklist. Inclusivity in global research.**
(DOCX)

## Acknowledgments

We are grateful to the study participants for sharing their thoughts and experiences with us, the research assistants for their commitment and dedication, and the staff in the CHEMCHA trial for their kind assistance.

## Author contributions

**Conceptualization:** Sarah Svege, Siri Lange, Bjarne Robberstad, Joseph Rujumba.

**Formal analysis:** Sarah Svege.

**Funding acquisition:** Sarah Svege, Bjarne Robberstad, Joseph Rujumba.

**Investigation:** Sarah Svege, Joseph Rujumba.

**Methodology:** Sarah Svege, Siri Lange, Bjarne Robberstad, Joseph Rujumba.

**Project administration:** Sarah Svege, Bjarne Robberstad, Joseph Rujumba.

**Supervision:** Siri Lange, Bjarne Robberstad, Joseph Rujumba.

**Writing – original draft:** Sarah Svege.

**Writing – review & editing:** Sarah Svege, Siri Lange, Bjarne Robberstad, Joseph Rujumba.

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
