## [Decision Letter · Decision Letter 0]

PGPH-D-24-00801

"I want to help my body": Acceptability of malaria chemoprevention among children with sickle cell anaemia and their caregivers in Malawi and Uganda

Dear Dr. Svege,

Thank you for submitting your manuscript to PLOS Global Public Health. After careful consideration, we feel that it has merit but does not fully meet PLOS Global Public Health’s publication criteria as it currently stands. Therefore, we invite you to submit a revised version of the manuscript that addresses the points raised during the review process.

We look forward to receiving your revised manuscript.

Kind regards,

Khadime Sylla

Academic Editor

Journal Requirements:

1. We ask that the manuscript source file is provided at Revision. Please upload your manuscript file as a .doc, .docx, .rtf or .tex.

2. Please provide your detailed Financial Disclosure statement. This is published with the article. It must therefore be completed in full sentences and contain the exact wording you wish to be published.

**Please only choose the relevant sentences from below**

a) Please clarify all sources of funding (financial or material support) for your study. List the grants (with grant number) or organizations (with url) that supported your study, including funding received from your institution.

b) State the initials, alongside each funding source, of each author to receive each grant.

c) State what role the funders took in the study. If funders had no role in your study, please state: "The funders had no role in study design, data collection and analysis, decision publish, or preparation of the manuscript."

d) If any authors received a salary from any of your funders, please state which authors and which funders.

Additional Editor Comments (if provided):

All remarks and comments from reviewers must be taken in considering the manuscript for publications.

This manuscript needs major revision in order to be considered for publication.

Reviewers' comments:

Reviewer's Responses to Questions

**Comments to the Author**

1. Does this manuscript meet PLOS Global Public Health’s publication criteria ? Is the manuscript technically sound, and do the data support the conclusions? The manuscript must describe methodologically and ethically rigorous research with conclusions that are appropriately drawn based on the data presented.

Reviewer #1: Yes

Reviewer #2: No

2. Has the statistical analysis been performed appropriately and rigorously?

Reviewer #1: Yes

Reviewer #2: N/A

3. Have the authors made all data underlying the findings in their manuscript fully available (please refer to the Data Availability Statement at the start of the manuscript PDF file)?

Reviewer #1: No

Reviewer #2: Yes

4. Is the manuscript presented in an intelligible fashion and written in standard English?

Reviewer #1: Yes

Reviewer #2: Yes

5. Review Comments to the Author

Reviewer #1: The Introduction

This is a well-written review of existing chemoprevention strategies for malaria in sickle cell disease. The limitations of some of these are highlighted, implying the need to search for others, and we believe this is the aim of the clinical trial of which this study is a part. However, we believe that the rationale for evaluating the acceptability or otherwise of a strategy by beneficiaries should be emphasised. This would better justify the need to always include qualitative studies in clinical trials. We think that a paragraph on this would have helped to frame the issue.

Methodology

We believe that this section should be rewritten in a simpler form for greater clarity, so that the reader has a better understanding of how the study was carried out.

We propose the following approach

- Study site and period (in this section, refer to the clinical trial protocol if it has been published; if not, describe the different areas with supporting maps and explain how and why the study areas were chosen).

- Study population (talk about the children and parents, specifying how they were chosen to take part in the focus groups; in short, talk about the sampling of participants included in this study).

- Type of study: talk here about the type of study, talking about the choice of three timelines (prospective acceptability, simultaneous acceptability and retrospective acceptability, with a definition of each concept).

- Conduct of the study: Explain how the focus groups were conducted, by whom and the supervisory activities of the research teams. It will be necessary to avoid naming individually what each author of the article did. You can do this in the section dedicated to the authors' roles.

The figures given in the data collection section (number of focus group participants, table 1) are, in our view, results and could be transferred to the results section.

- Data Analysis: This section does not need to be changed, just state that the quantitative analysis (description of participants) was carried out and describe how, not forgetting the analysis software used.

-Ethical considerations: This section is very well written, but it would be useful to describe how the consent forms were managed to ensure confidentiality, as this was a multicentre study. This means stating whether each country kept its own forms or whether they were centralized.

The results are well described and the discussion is well conducted. We would also like to congratulate the authors on their research efforts, which are reflected in the number of references used.

Reviewer #2: This acceptability study should have been presented after the efficacy study. It would be preferable to discuss the efficacy and safety of chemoprevention first. Giving PD on a weekly basis seems too burdensome when we know that PD has a long shelf-life, especially piperaquine (more than 20 days in the blood). Secondly, the study makes no mention of a first dose taken under TDO, which is generally de rigueur when taking these long-lasting drugs.

Also, this study lacks a contextual analysis. There is no reminder of the epidemiology of malaria in Malawi and Uganda to understand the periodicity of chemoprevention administration. For whenever chemoprevention is used, it responds to the duration of the malaria transmission season, but not on an annual basis, as is the case here. The article does not mention whether CMS is implemented in these two countries, as the age range of the children in the study includes a large proportion of the CMS target group (6 months to 5 years).

6. PLOS authors have the option to publish the peer review history of their article (what does this mean? ). If published, this will include your full peer review and any attached files.

**Do you want your identity to be public for this peer review?** For information about this choice, including consent withdrawal, please see our Privacy Policy .

Reviewer #1: No

Reviewer #2: No

---

## [Decision Letter · Decision Letter 1]

PGPH-D-24-00801R1

"I want to help my body": Acceptability of malaria chemoprevention among children with sickle cell anaemia and their caregivers in Malawi and Uganda

Dear Dr. Svege,

Thank you for submitting your manuscript to PLOS Global Public Health. After careful consideration, we feel that it has merit but does not fully meet PLOS Global Public Health’s publication criteria as it currently stands. Therefore, we invite you to submit a revised version of the manuscript that addresses the points raised during the review process.

We look forward to receiving your revised manuscript.

Kind regards,

Miquel Vall-llosera Camps

Staff Editor

Journal Requirements:

Reviewers' comments:

Reviewer's Responses to Questions

**Comments to the Author**

1. If the authors have adequately addressed your comments raised in a previous round of review and you feel that this manuscript is now acceptable for publication, you may indicate that here to bypass the “Comments to the Author” section, enter your conflict of interest statement in the “Confidential to Editor” section, and submit your "Accept" recommendation.

Reviewer #1: All comments have been addressed

Reviewer #3: (No Response)

2. Does this manuscript meet PLOS Global Public Health’s publication criteria ? Is the manuscript technically sound, and do the data support the conclusions? The manuscript must describe methodologically and ethically rigorous research with conclusions that are appropriately drawn based on the data presented.

Reviewer #1: Yes

Reviewer #3: Yes

3. Has the statistical analysis been performed appropriately and rigorously?

Reviewer #1: Yes

Reviewer #3: N/A

4. Have the authors made all data underlying the findings in their manuscript fully available (please refer to the Data Availability Statement at the start of the manuscript PDF file)?

Reviewer #1: Yes

Reviewer #3: Yes

5. Is the manuscript presented in an intelligible fashion and written in standard English?

Reviewer #1: Yes

Reviewer #3: Yes

6. Review Comments to the Author

Reviewer #1: Thank for taking in account all comments

Reviewer #3: 156-165: Use data from the latest WHO 2024 report.

320: had completed 5-7 grades of schooling (51.4%). This information is not found in Table 2.

336. Please review the way of presenting the tables should have only three horizontal lines in its presentation. Two lines to frame the labels and one line at the end of the table.

In Table 3, do you talk about the number of children with SCD or SCA?

382-385: Please provide a verbatim statement from your survey to argue the fact that "a few participants argued that weekly drugs may provide better protection against..."

386-396: Please review the title and content. Here, the verbatim statement you provided mainly indicates the difficulty of going to the hospital to get the medication.

569-571: “To our knowledge, this is the first study assessing acceptability towards weekly single-day treatment courses of DP. Weekly dosing of DP is predicted to be more effective at preventing malaria episodes than monthly dosing…” In your results section you should provide more evidence to support what you are saying here.

575-577: “In the final phase of data collection, we recruited a sub-sample of participants who had received weekly DP for an average period of 15 months as part of the parent trial. The retrospective phase was performed after trial finalisation and unbinding.” We did not feel in the presentation of your results a presentation according to the different phases of your data collection.

7. PLOS authors have the option to publish the peer review history of their article (what does this mean? ). If published, this will include your full peer review and any attached files.

**Do you want your identity to be public for this peer review?** For information about this choice, including consent withdrawal, please see our Privacy Policy .

Reviewer #1: No

Reviewer #3: No

---

## [Decision Letter · Decision Letter 2]

"I want to help my body": Acceptability of malaria chemoprevention among children with sickle cell anaemia and their caregivers in Malawi and Uganda

PGPH-D-24-00801R2

Dear Ms. Svege,

We are pleased to inform you that your manuscript '"I want to help my body": Acceptability of malaria chemoprevention among children with sickle cell anaemia and their caregivers in Malawi and Uganda' has been provisionally accepted for publication in PLOS Global Public Health.

Best regards,

Julia Robinson

Executive Editor

Reviewer Comments (if any, and for reference):

Reviewer's Responses to Questions

**Comments to the Author**

1. If the authors have adequately addressed your comments raised in a previous round of review and you feel that this manuscript is now acceptable for publication, you may indicate that here to bypass the “Comments to the Author” section, enter your conflict of interest statement in the “Confidential to Editor” section, and submit your "Accept" recommendation.

Reviewer #1: All comments have been addressed

Reviewer #3: All comments have been addressed

2. Does this manuscript meet PLOS Global Public Health’s publication criteria ? Is the manuscript technically sound, and do the data support the conclusions? The manuscript must describe methodologically and ethically rigorous research with conclusions that are appropriately drawn based on the data presented.

Reviewer #1: Yes

Reviewer #3: Yes

3. Has the statistical analysis been performed appropriately and rigorously?

Reviewer #1: Yes

Reviewer #3: Yes

4. Have the authors made all data underlying the findings in their manuscript fully available (please refer to the Data Availability Statement at the start of the manuscript PDF file)?

Reviewer #1: No

Reviewer #3: Yes

5. Is the manuscript presented in an intelligible fashion and written in standard English?

Reviewer #1: Yes

Reviewer #3: Yes

6. Review Comments to the Author

Reviewer #1: (No Response)

Reviewer #3: All comments have been addressed

7. PLOS authors have the option to publish the peer review history of their article (what does this mean? ). If published, this will include your full peer review and any attached files.

**Do you want your identity to be public for this peer review?** For information about this choice, including consent withdrawal, please see our Privacy Policy .

Reviewer #1: No

Reviewer #3: No
